# Transfer Learning on Manifolds via Learned Transport Operators

## Abstract

Within-class variation in a high-dimensional dataset can be modeled as being on a low-dimensional manifold due to the constraints of the physical processes producing that variation (e.g., translation, illumination, etc.). We desire a method for learning a representation of the manifolds induced by identity-preserving transformations that can be used to increase robustness, reduce the training burden, and encourage interpretability in machine learning tasks. In particular, what is needed is a representation of the transformation manifold that can robustly capture the shape of the manifold from the input data, generate new points on the manifold, and extend transformations outside of the training domain without significantly increasing the error. Previous work has proposed algorithms to efficiently learn analytic operators (called transport operators) that define the process of transporting one data point on a manifold to another. The main contribution of this paper is to define two transfer learning methods that use this generative manifold representation to learn natural transformations and incorporate them into new data. The first method uses this representation in a novel randomized approach to transfer learning that employs the learned generative model to map out unseen regions of the data space. These results are shown through demonstrations of transfer learning in a data augmentation task for few-shot image classification. The second method use of transport operators for injecting specific transformations into new data examples which allows for realistic image animation and informed data augmentation. These results are shown on stylized constructions using the classic swiss roll data structure and in demonstrations of transfer learning in a data augmentation task for few-shot image classification. We also propose the use of transport operators for injecting transformations into new data examples which allows for realistic image animation.

## 1 Introduction

While significant progress has been made in training classifiers that can effectively discriminate between thousands of classes, the increasing classifier complexity obfuscates the reasoning behind class selection and requires large training datasets to capture variations within each class. In many settings, the within-class variation in a high-dimensional dataset can be modeled as being low-dimensional due to the constraints of the physical processes producing that variation (e.g., translation, illumination, etc.). When these variations are within a linear subspace, classic techniques such as principal component analysis (PCA) can be used to efficiently capture the transformations within the data. However, the manifold hypothesis states that in many cases of interest (e.g., images, sounds, text), the within-class variations lie on or near a low-dimensional nonlinear manifold (Bengio et al., 2013). In the neuroscience literature there is a hypothesis that the manifold nature of these variations is explicitly exploited by hierarchical processing stages to untangle the representations of different objects undergoing the same physical transformations (e.g., pose) (DiCarlo & Cox, 2007).

By learning manifold representations of identity-preserving transformations on a subset of classes or data points, we gain the ability to transfer these natural variations to previously unseen data. A generative model of the transformation manifold trained on a dataset rich in variation can be used to transfer knowledge of those variations to previously unseen domains. This can increase robustness and reduce the training burden for machine learning tasks as well as enable the generation of novel examples. Additionally, an explicit understanding of the transformations occuring within a dataset

provides interpretibility of machine learning tasks that is typically unavailable. One can view this approach as a variant of pattern theory that seeks transformations that operate on representational primitives (Grenander, 1994), but with an explicit unsupervised learning of the low-dimensional manifold structure of many real-world datasets.

There are a large number of "manifold learning" algorithms that have been introduced in the literature to discover manifold structure from data. The most common approach to this task is to perform an embedding of the original data points after performing a transformation to preserve either local or global properties of the manifold (e.g., local neighborhood relationships, global geodesic distances, etc.) (Tenenbaum et al., 2000; Roweis & Saul, 2000; Weinberger & Saul, 2006; Belkin & Niyogi, 2003). Unfortunately, such approaches capture manifold structure through a transformation of the data points themselves into a lower dimensional space and therefore are not suitable for the desired tasks. Specifically, there is no generative model of the data in the original high-dimensional space, meaning that the inferred manifold structure is not transferable to other data classes, is not amenable to strong interpolation/extrapolation, and does not provide a probabilistic model that can be used in machine learning tasks. More recently, there there are a number of methods that have been introduced that capture manifold structure through a variety of approaches that involve estimating local tangent planes (Dollár et al., 2007b;a; Bengio & Monperrus, 2005; Park et al., 2015). While these methods can admit representations that have some of the above advantages of generative models, the linear approximations can cause challenges when trying to perform transfer learning by extrapolating to out-of-sample points in manifold locations not well-represented in the training data.

As an alternative to the above, previous work has proposed unsupervised learning algorithms to efficiently learn Lie group operators that capture the structure of low-dimensional manifolds (Culpepper & Olshausen, 2009; Van Gool et al., 1995; Miao & Rao, 2007; Rao & Ruderman, 1999). The manifold representation that is described by analytic operators (called *transport operators*) defines the process of transporting one data point to another, thereby providing a probabilistic generative model for the manifold structure (Culpepper & Olshausen, 2009). The main contribution of this paper is to define two transfer learning methods that use this generative manifold representation to learn natural transformations and incorporate them into new data. The first method uses this representation in a novel randomized approach to transfer learning that employs the learned generative model to map out unseen regions of the data space. These results are shown through demonstrations of transfer learning in a data augmentation task for few-shot image classification. The second method use of transport operators for injecting specific transformations into new data examples which allows for realistic image animation and informed data augmentation.

## 2 BACKGROUND AND RELATED WORK

Transfer learning problems can take a variety of forms (Pan & Yang, 2010), including using training data to approximate target distributions (after modification) (Shimodaira, 2000; Wang & Schneider, 2014; Huang et al., 2006; Gong et al., 2012; Ben-David et al., 2010), bias a classifier (Eaton et al., 2008), or define a prior on parameters (Raina et al., 2006; Srivastava & Salakhutdinov, 2013). Research by Freifeld et al. (2014) introduced a framework that can transfer a data distribution from one area of a manifold to another using parallel transport of the model parameters along the manifold. Some data augmentation techniques are closely related to transfer learning because they attempt to expand limited training sets beyond the original domain using information from classes with many examples (Wei et al., 2012; Hauberg et al., 2015; Freifeld et al., 2015).

Many of the manifold learning techniques for non-linear manifolds (e.g., Isomap (Tenenbaum et al., 2000), Locally-Linear Embedding (LLE) (Roweis & Saul, 2000), Maximum Variance Unfolding (MVU) (Weinberger & Saul, 2006), and Laplacian Eigenmaps (Belkin & Niyogi, 2003)) represent the manifold through a low-dimensional embedding of the data points. Although extensions to out-of-sample points have been proposed for several manifold learning techniques, these are highly dependent on the initial domain of the training data (Bengio et al., 2004). In contrast, there have been several recent techniques introduced (e.g., Locally Smooth Manifold Learning (LSML) (Dollár et al., 2007b;a) and Non-Local Manifold Tangent Learning (Bengio & Monperrus, 2005)) that use the data points to learn a function that maps high dimensional points to tangent planes to represent the manifold. While these tangent planes can be estimated anywhere in the data space, error can accumulate quickly in the linear approximations when extrapolated away from the training domain.

Figure 1: Trajectories of the two dictionary elements, $\Psi_m$, trained on point pairs on a swiss roll. Each plot shows a single transport operator acting on several example points.

The recent Locally Linear Latent Variable Model (LL-LVM) estimates both the embedded points and the mapping between the high-dimensional and low-dimensional data using a probabilistic model which tries to maximize the likelihood of the observations (Park et al., 2015). While similar in spirit to LSML and Non-Local Manifold Tangent Learning, this technique has the added benefit of a metric which determines the quality of an embedding and a method to reconstruct high-dimensional representations of out-of-sample points in the embedding.

## 3    LEARNING ACCURATE AND ROBUST MANIFOLD REPRESENTATION

Manifold transport operators are Lie group operators that capture the paths on a manifold between data points through an analytic operator (Culpepper & Olshausen, 2009). In this approach, we assume that two nearby points $\mathbf{x}_0, \mathbf{x}_1 \in \mathbb{R}^N$ living on a low-dimensional manifold can be related through a dynamical system that has the solution path

$$\mathbf{x}_1 = \mathrm{expm}(\mathbf{A})\mathbf{x}_0 + \mathbf{n}, \tag{1}$$

where $\mathbf{A} \in \mathbb{R}^{N \times N}$: $\dot{\mathbf{x}} = \mathbf{A}\mathbf{x}$ is the operator capturing the dynamics, $\mathbf{n}$ is the error, and $\mathrm{expm}$ is a matrix exponential. To allow for different geometrical characteristics at various points on the manifold, each pair of points may require a different dynamics matrix $\mathbf{A}$. We assume that this matrix can be decomposed as a weighted sum of $M$ dictionary elements ($\mathbf{\Psi}_m \in \mathbb{R}^{N \times N}$) called *transport operators*:

$$\mathbf{A} = \sum_{m=1}^{M} \mathbf{\Psi}_m c_m. \tag{2}$$

The transport operators represent a set of primitives that describe local manifold characteristics. For each pair of points (i.e., at each manifold location), the geometry will be governed by a small subset of these operators through the weighting coefficients $\mathbf{c} \in \mathbb{R}^M$.

Using the relationship between points in (1) and the decomposition in (2), we can write a probabilistic generative model that allows efficient inference. We assume a Gaussian noise model, a Gaussian prior on the transport operators (model selection), and a sparsity inducing prior on the coefficients (model regularization). The resulting negative log posterior for the model is given by

$$\frac{1}{2} \left\| \mathbf{x}_1 - \mathrm{expm}\left(\sum_{m=1}^{M} \mathbf{\Psi}_m c_m\right) \mathbf{x}_0 \right\|_2^2 + \frac{\gamma}{2} \sum_m \|\mathbf{\Psi}_m\|_F^2 + \zeta\|\mathbf{c}\|_1 \tag{3}$$

where $\|\cdot\|_F$ is the Frobenius norm. Following the unsupervised algorithm in (Culpepper & Olshausen, 2009), we use pairs of nearby training points to perform unsupervised learning on the transport operators using descent techniques (alternating between the coefficients and the transport operators) on the objective in (3).

Using the swiss roll manifold as a benchmark, we learn transport operators by randomly sampling points in a defined area on the manifold. During training, we randomly select an initial point ($\mathbf{x}_0$) and select a second point ($\mathbf{x}_1$) from a defined neighborhood surrounding $\mathbf{x}_0$. For this pair we infer the coefficients $\mathbf{c}$ as described above using a fixed dictionary $\{\mathbf{\Psi}_m\}$. Using a batch of these point

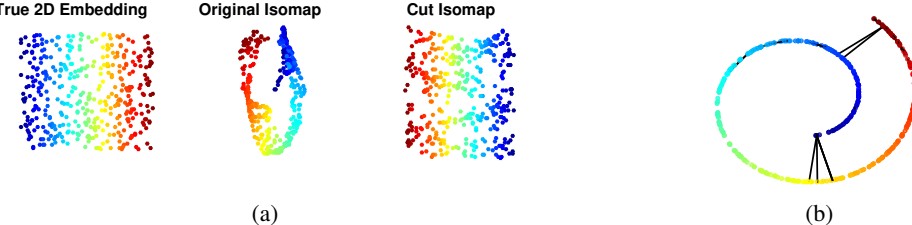

Figure 2: (a) Left column: True embedding of 400 swiss roll points. Middle column: Isomap embedding of swiss roll points for neighborhoods defined by k-nearest neighbors with $k = 11$. The embedding is distorted due to the large neighborhoods which include shortcuts. Right column: Embeddings after the transport operator objective function is used to identify and eliminate shortcuts. (b) View of the swiss roll points with lines between the points which had connections broken due to large objective function values.

pairs and associated coefficients we then take a gradient step on the objective with respect to the dictionary while holding the coefficients fixed. In this example we fix the number of dictionary elements to two (setting $\gamma = 0$) and set $\zeta = 2$.

Fig 1 shows the trajectories of the transport operator dictionary elements resulting from this training. Each plot uses a few example starting points on the swiss roll, $\mathbf{x}_i(0)$, and visualizes the effect of a single transport operator by applying a single dictionary element $\Psi_m$ to each point in time steps $\mathbf{x}_i(t) = \exp(\Psi_m t)\mathbf{x}_i(0)$, $t = 0, ...., T$. In this case, the two operators have learned to move in an expanding (shrinking) rotational motion and in a transverse motion.

With the objective function as a similarity measure between pairs of points and running classical Multidimensional scaling (MDS), we can create an embedding of the points in 2D that shows their intrinsic low-dimensional structure. While many of the classic manifold learning techniques mentioned earlier are designed specifically to find such embeddings, they can be fragile to algorithmic assumptions. For example, most other manifold learning techniques require a local neighborhood definition and even small mistakes in the neighborhood definition can lead to major errors if shortcuts are created (i.e., points nearby in Euclidean distance are considered neighbors even though they are far away on the manifold). Additionally, with many algorithms there is no way to know if a neighborhood definition has caused a shortcut error to occur, and therefore there is no way to assess the quality of the output embedding.

In contrast, the representation learned through the transport operator approach shows significant robustness properties due to two factors. While the transport operator model does require a neighborhood definition to select point pairs during training, this algorithm is more robust to mistakes in this definition because the information from each pair is averaged with many other point pairs in the learning process. Furthermore, after learning the transport operators on the dataset, the objective function value in (3) can be used to evaluate the likelihood of points being in the same neighborhood on the low-dimensional manifold.

Fig 2 provides an example how the objective function metric can be used to correct the neighborhoods for an isomap embedding (Tenenbaum et al., 2000). When the isomap embedding is computed with a neighborhood defined by the k-nearest neighbors with $k = 11$, there are shortcut connections that lead to a distorted embedding. We compute the objective function between each pair of points using the learned transport operator representation and "cut" the connections shown in Fig 2(b) where the objective function is greater than threshold (based on outliers in a histogram of the objective function values). Running isomap on the corrected neighborhood definition produces a much more accurate embedding as demonstrated in Fig 2(a). As a note, while the recently proposed LL-LVM (Park et al., 2015) defines a probabilistic model that can be similarly used to identify when an entire embedding is bad due to a neighborhood error, the transport operator approach defines an objective function on each point pair that can be used to identify precisely which pairs are causing erroneous neighborhood definitions.

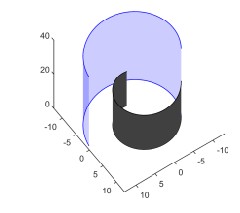 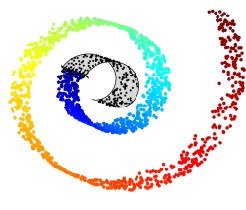

(a) Transfer learning scenario     (b) Manifold extrapolation

Figure 3: (a) The dark gray shaded area shows the training domain and the blue shaded area shows the testing domain. (b) The transport operators trained using only the black points in the gray shaded region and are applied with random sets of coefficients to the black 'x' points at the edge of the training domain to extrapolate to points outward.

## 4 TRANSFER LEARNING

As mentioned earlier, the probabilistic generative model underlying the transport operators approach allows for the possibility of generating new points on (or near) the manifold that are consistent with the manifold model of natural variations. This model can be used to explore and expand the manifolds associated with new data points by transferring the model variations. In order to succeed in transferring transformations, we must be able to interpolate paths between two points that remain on or near the manifold and extrapolate realistic transformations beyond the training domain. Classic techniques based on data embeddings are hindered in their ability to interpolate between points because they are limited to simple interpolation techniques between existing data points in the embedding space. Techniques that learn global manifold mappings (i.e., LSML and Non-Local Manifold Tangent Learning) can similarly create paths between points by using the mapped tangent planes and a snake active contour method (see details in (Dollár et al., 2007b)).

We propose a novel transfer learning method in two scenarios that both utilize the generative model defined by the transport operator framework. In the first method we randomly apply learned manifold transformations to map out unseen regions of the manifold. With a single example data point from the target distribution, we can use small magnitude random coefficients to explore the data space with a model consistent with the training dataset on another portion of the manifold. With no examples from the target area of the manifold, we can use larger magnitude random coefficients to map out unseen portions of the space with paths that are consistent with the observed manifold geometry. In the second method, we transfer specific manifold variations from the training dataset to new example points by precomputing structured coefficients associated with transformations of interest. In each case, to be successfully used for transfer learning, the manifold transformations must remain faithful representations of the manifold shape outside the original training domain. We illustrate our approaches to transfer learning on the swiss roll manifold (where we have ground truth) and then provide examples of both transfer learning methods on the USPS handwritten digits and facial expression databases.

We first demonstrate transfer learning on a swiss roll manifold by training transport operators on one limited portion of the swiss roll that does not represent the behavior of the manifold as a whole. Specifically, the training set does not represent the range of tangent plane orientations of the full model, and it does not extend the full width of the transverse directions present. Figure 3a shows the training domain in dark gray. The testing region is shown in blue, and it has no overlap with the training region. As described earlier, we can use the generative properties of the transport operator model to transfer our learned manifold knowledge by extrapolating from the training region to the testing region. Specifically, we extrapolate to new data points outside the training region by using data points on the edge of the training region as the starting point and applying the learned transport operators with random coefficients sampled from a uniform distribution. Repeating that process many times, Fig. 3b shows how we can transfer the learned knowledge to map out an approximation of the new portion of the swiss roll manifold. In this demonstration, each colored dot is the result of a random extrapolation from a datapoint on the edge of the training region using a single randomized set of coefficients. This provides an example of random manifold mapping required for the first transfer learning method.

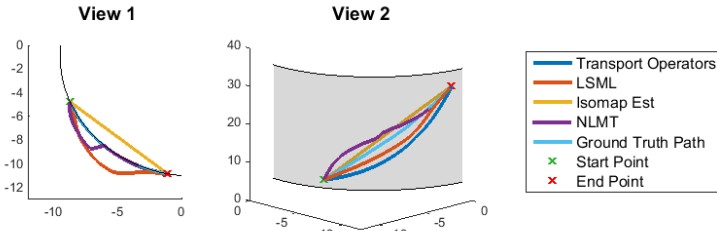

Figure 4: Example estimates of paths on the manifold between points from the testing set in a transfer learning setting. The transport operator path follows the manifold much more closely than path estimates from the other algorithms. Ground truth is the geodesic path between the points.

The other techniques that learn global manifold mappings (i.e., LSML, Non-Local Manifold Tangent Learning) can extrapolate by computing local tangent directions at each point and projecting from known points along those directions. Because of the linearity assumption inherent in tangent plane approximations, there are limits to the size of the extrapolations that are likely before departing from the manifold. We demonstrate this phenomenon by testing the ability to estimate paths between points outside of the training domain. Using the same training/testing regions as before, we select two points from the testing region (where no training data has been seen) and use the various manifold representations to infer a path on the manifold between the two points. Fig 4 shows example paths between points selected from the testing domain. In this case, the transport operator path is the only one that does not depart significantly from the manifold. Because the LSML and Non-Local Manifold Tangent Learning paths are estimated by recomputing tangents along the route to ensure the path remains on the manifold, the path offset suggests an error in the tangent mappings in this area due to the differences between the training and testing regions.

To quantify the error in the manifold representation when transferring from the training region to the testing region, as above we select point pairs and use each method to find paths between the points that are near the manifold. We compute a path offset metric that indicates how far an estimated path departs from the true manifold by first calculating the maximum distance between any point on the path and its closest corresponding point on a densely sampled manifold. The manifold points are defined using a uniformly spaced grid over a swiss roll with a distance of 0.4 between neighboring points. The distance metric is the mean value of the maximum offsets for all paths in the test set. We evaluate each algorithm using point pairs from the training region (Fig 5a), and in the transfer learning scenario, using point pairs from the testing region (Fig 5b). When no transfer is required, all algorithms are able to generate paths with small maximum distances to the manifold. While overall performance decreases for all algorithms in the transfer learning case, the transport operator approach is consistently able to produce paths with smaller deviations from the manifold. In the transfer setting, we explore the performance on this task in more detail by dividing the test cases up into distance categories based on the ground truth path length. Fig 5c shows the performance as a function of distance category when the methods are trained with 300 points, illustrating that the transport operator approach is performing notably better for longer path lengths. The ability to map out longer paths that remain close to the manifold highlights the benefit of using transport operators for defining and transfering paths that represent specific transformations.

## 4.1 RANDOM MANIFOLD MAPPING USING USPS DATASET

We utilize the USPS handwritten digit image dataset (Hull, 1994) to demonstrate our ability to use transport operator representations to perform transfer learning by randomly mapping the transformation manifold. We can explore this in detail through a few-shot classification task. For training, we create a dataset that consists of 1000 examples of the digit '8' paired with that same image rotated $2.5°$ and transport operators are learned between those point pairs.[1] We define the neighborhood graph with one nearest neighbor and all techniques are parametrized to learn a one-dimensional manifold (i.e. $M = 1$ and $\gamma = 0$). In other words, without telling the algorithms about the concept

---

[1]The images are padded to $18 \times 18$ to allow rotation and cropping without losing digit information.

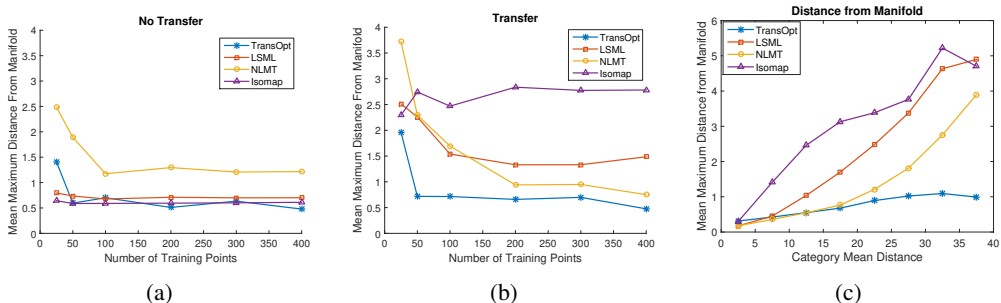

(a)            (b)            (c)

Figure 5: Path deviation from the manifold as a function of number of training points when the point pairs (a) are chosen from the training domain or (b) in a transfer setting when point pairs are selected from the testing domain which is distinct from the training domain. (c) Path deviation in the transfer learning task as a function of true path distance.

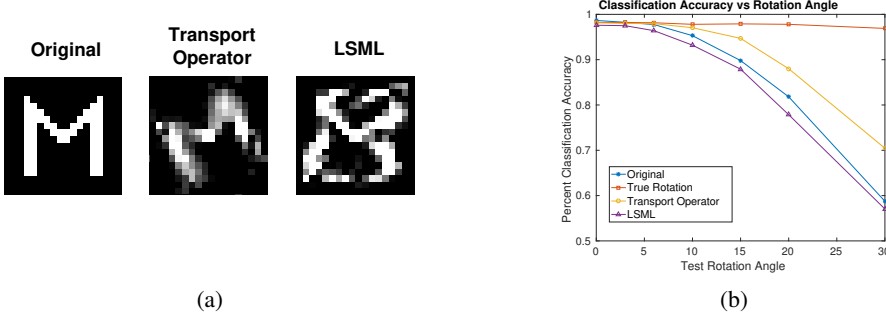

(a)                          (b)

Figure 6: (a) An example of manifold transformations trained on '8' digits being applied to the letter 'M'. (b) Convolutional neural network classifiers are tested on rotated USPS digits. The classifiers are trained on USPS digits in four variations: the original data, data trained with true rotations of $3, 6, 10, 30°$, and data augmented through transfer learning using transport operators and LSML manifold approximations based on only seeing rotated '8' digits.

of rotation, we seek to have them learn the general transformation manifold from examples of only slightly rotated '8' digits. This task extends the transfer learning example described in (Bengio & Monperrus, 2005) where they trained the Non-Local Manifold Tangent Learning mapping on all numerical digits and tested on a letter. In this case, we are training only on a single digit, providing less information for learning and increasing the chance of overfitting to the training class.

To highlight the performance when information is transferred between manifolds, we apply the transformation learned on rotated '8' digits to the letter 'M'. Fig. 6a shows the original 'M' as well as the result after applying the learned transformation from the transport operator approach and LSML. Despite being trained only on slightly rotated '8' digits, the transport operator can rotate the 'M' by nearly $45°$ while maintaining the shape of the letter and without inducing much distortion. When the LSML tangent is applied to an 'M', it transforms the M to look more like an '8'. This result indicates that the tangent mapping has learned a transformation that is more specific to a rotated '8' digit and it cannot be easily applied in this transfer setting to other inputs.

One application of transfer learning is data augmentation for few-shot learning classification tasks. In this approach, an impoverished training set from one class is bolstered by creating surrogate training data by transfer learning from another class with more abundant training data. In other words, the desire is to perform few-shot learning on new classes by transferring the variability learned from other classes into an augmented dataset for the new class. We test this data augmentation technique on rotated USPS digits classified by a convolutional neural network classifier (LeCun et al., 1998).[2] In this demonstration we use four versions of the classifier that are tested using different training sets: 1) only the original USPS digits with no rotation introduced (naive classification), 2) the orig-

---

[2]We employed a LeNet convolutional network with $5 \times 5$ filters in the first and second convolutional layers.

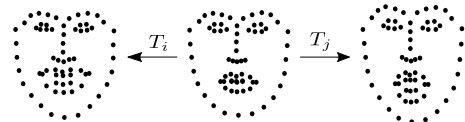

Figure 7: The center image shows the neutral facial landmark points. The left column shows the result of applying a single transport operator that qualitatively creates "happiness" and the right column shows the result of applying a transport operator that qualitatively creates "surprise."

inal USPS digits each rotated by 3, 6, 10, and 30 degrees (oracle classification), 3) the USPS digits transformed many times by using random coefficients with the transport operator that was learned only on rotated '8' digits, and 4) the USPS digits transformed by the LSML tangent vector learned only on rotated '8' digits.

We test the classification accuracy on USPS digits rotated by varying angles and the resulting accuracies are shown in Fig 6b. As expected, the network trained with the rotated digits (oracle classification) is not significantly affected by a change in the rotation angle of the test digit and the network trained only on the original USPS data with no rotation (naive classification) experiences a significant performance decrease with larger rotations. The network trained using an augmented dataset from transfer learning with transport operators improves the performance significantly over naive classification. However, due to the distortions present with large transformations using tangent space approximations, the network trained using an augmented dataset from transfer learning with LSML achieves no performance increase (and may even see a performance deficit) compared to the naive classification.

## 4.2 STRUCTURED MANIFOLD TRANSFER USING FACIAL EXPRESSION DATASETS

We employ facial expression datasets to demonstrate the second transfer learning method which uses transport operators to generate transformations in new settings by transferring a structured set of coefficients. Both the MUG facial expression database (Aifanti et al., 2010) and the Extended Cohn-Kanade (CK+) database (Kanade et al., 2000; Lucey et al., 2010) contain image sequences from a variety of subjects making six expressions (anger, disgust, fear, happiness, sadness, and surprise). Each sequence starts with a neutral face and reaches the apex of the expression. By learning transport operators on landmark points from faces in these sequences, we can identify natural facial dynamics that lead to expressions.

The transport operators are trained on the facial landmark points from pairs of images in the MUG database expression sequences. Only 12 of the 52 subjects in this database are used for transport operator learning. We identify landmark points in each image using the facial landmark detection functions in the dlib python library which detect 68 points of interest on a given face. We swept over the parameters in (3) and found the best performance for $\gamma = 5 \cdot 10^{-5}$, $\zeta = 0.01$, and $M = 20$. Because the training is unsupervised, there is not a single transport operator associated with each expression. However, several operators elicit transformations that can be qualitatively associated with expressions (see Fig 7 for two examples).

The learned transport operators can be used to generate realistic expression sequences for new subjects. We interpolate paths between two points outside of the original training domain using both transport operators and LSML as we did with the swiss roll in section **??**. We generate the paths between the landmarks in the first (neutral) and last image (expression apex) in the expression sequences in the CK+ database. We compute the same path offset metric as in section **??** using the real expression sequence as the ground truth manifold and defining the error as the distance between landmark points in the ground truth frames and the generated frames. The mean maximum path offset for all the paths in the CK+ database is 0.5526 for transport operators and 0.5978 for LSML.

We can also use transport operators to extrapolate an expression from neutral landmark points. This requires transformations that can be applied to landmarks from neutral faces to create each expression. In the transport operator setting, a specific transformation is defined by a set of coefficients that control the learned transport operators as in (2). Prior to extrapolating expressions from neutral landmark points, we need to compute the coefficients for each desired expression. Coefficient

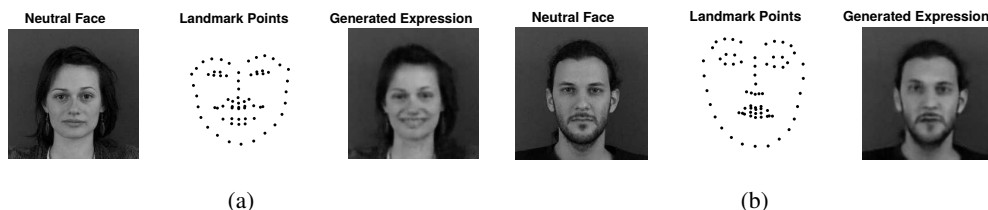

(a)                                                                                              (b)

Figure 8: Examples of images with generated expressions. The neutral image (left column) and the landmark points (middle column) are input into a conditional GAN which outputs the generated face with the expression specified by the landmark points (right column). (a) Example of happiness expression generation. (b) Example of sadness expression generation

inference requires landmarks from an image at the apex of the expression ($\mathbf{x}_1$) and landmarks from a neutral image in the same sequence ($\mathbf{x}_0$). Once the coefficients are obtained for expression $i$, we have a dynamics matrix, $\mathbf{A}_i = \sum_{m=1}^{M} \mathbf{\Psi}_m c_{i,m}$, the can be applied to any set of neutral landmark points to create expression $i$.

To show the power of this expression generation, we incorporate it into a generative adversarial network (GAN) (Goodfellow et al., 2014). We implement a conditional GAN (Isola et al., 2016; Wang & Gupta, 2016; Makhzani et al., 2015) in tensorflow that trains a generator network that is given a neutral image conditioned on a set of landmark points and outputs the person from the neutral image making the expression specified by the landmarks points. This conditional GAN is trained on images and facial landmark points from the MUG database. To generate a new expression image, we select a neutral image, apply the dynamics matrix $\mathbf{A}_i$ to the landmark points from that image, and input the image and the landmarks into the generator. Figure 7 shows the result of adding an expression to two subject from the MUG database who were not included in the transport operator training set.

Expression extrapolation can be used for several applications of data generation. Using the conditional GAN, this technique could allow for natural facial animation where we transfer expressions or other facial changes to an individual from only one input point. Additionally, we can exploit natural transformations learned from unlabeled data to augment another dataset with limited labeled data to improve classification.

We test this structured data augmentation by using the transport operators trained on the MUG database to augment a small amount of labeled data from the CK+ database. We assume the only labeled data for each expression is one neutral image and an associated expression image from the same sequence. In this one-shot training scenario, we randomly choose one example sequence for each of the six expressions and infer the coefficients between landmarks in the first frame (neutral) and the last frame (expression apex). These coefficients are then applied to the neutral frames from all the remaining sequences in the training set. A support vector machine (SVM) is trained with this generated expression data to classify the six expressions as well as the neutral pose. Figure 9 shows the classification accuracy for 500 trials. When training with only single examples of each expression, the average accuracy is 0.4787. The transport operator data augmentation improved the average accuracy to 0.643. This shows the benefit of using transport operators to transfer information from an unsupervised dataset to a classification scenario with limited training samples.

## 5  CONCLUSION

We have shown that we can transfer meaninful transformation information between classes and examples using manifold transport operators which provide accurate and robust characterization of manifold data in the context of a probabilistic generative model. We have demonstrated that the learned transformations can be transferred accurately to other portions of the manifold (including out-of-sample extensions) through applying the generative model with both randomized and structured coefficients. The transfer learning potential was shown in the context of data generation and augmentation applications by animating individuals with new facial expression and providing examples of few-shot learning on digit and facial expression classification. These results constitute some

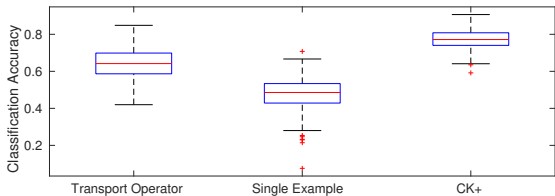

Figure 9: The boxes represent the classification accuracy distribution when the classifier is trained using the transport operator-augmented dataset (transport operator), using only one example per expression without data augmentation (single example), and using landmarks from the expression frames in all of the training sequences from the CK+ database (CK+).

of the first demonstrations that transport operators can form the basis of a learned representation for manifold data that has utility in applications and transfer learning tasks. The presented simulations are proof-of-concept simulations that allow us to fully explore and visualize the results in a way that is impossible with larger or more complex datasets.

While successful in these demonstrations, as with any algorithm the results depend on the data characteristics being sufficiently captured by the model family. Though there is evidence in the literature that Lie group operators can capture complex transformations in images (Culpepper & Olshausen, 2009), it is unknown if the transport operator approach will be sufficient to represent the rich variations in more complex datasets. Future work will be required to determine the complexity of manifold models that can be captured successfully by the proposed manifold transport operator approach, including demonstrations of similar tasks on more complex image classification tasks. Additionally, the current model only captures local transformations between points, but it will likely be beneficial to develop the model further to capture more regularity in the coefficient behavior across multiple pairs of points.

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
