# OpenReview forum: "Transfer Learning on Manifolds via Learned Transport Operators"
_ICLR.cc/2018/Conference — Reject_

### Official Review · AnonReviewer3 · 2017-11-20
**Fun idea, limited experiments, unclear intuition of proposed dynamics**

**Rating:** 4
**Confidence:** 4

**Review:**

Overview:
The paper aim to model non-linear, intrinsically low-dimensional structure, in data by estimating "transport operators" that predict how points move along the manifold. This is an old idea, and the stated contribution of the paper is:
"The main contribution of this paper is to show that the manifold representation learned in the transport operators is valuable both as a probabilistic model to improve general machine learning tasks as well as for performing transfer learning in classification tasks."
The paper provide nice illustrative experiments arguing why transport operators may be a useful modeling tool, but does not go beyond illustrative experiments.
While I follow the intuitions behind transport operators I am doubtful if they will generalize beyond very simple manifold structures (see detailed comments below).

Quality:
The paper is well-written and fairly easy to follow. In particular, I appreciate that the authors make no attempt to overclaim contributions. From a methodology point-of-view, the paper has limited novelty (transport operators, and learning thereof has been studied elsewhere), but there are some technical insights (likelihood model, use in data augmentation). Since the provided experiments are mostly illustrations, I would argue that the significance of the paper is limited. I'd say that to really convince a broader audience that this old idea is worth revisiting, the work must go beyond illustrations and apply to a real data problem.

Detailed Comments and Questions:
*) Equation 1 of the paper describe the key dynamics of the applied transport operators. Basically, the paper assume that the underlying data manifold is locally governed by a linear differential equation. This is a very suitable assumption, e.g., for the swiss roll data set, but it is unclear to this reader why it is a suitable assumption beyond such toy data. I would very much appreciate a detailed discussion of when this is a suitable modeling choice, and when it is not. My intuition is that this is mostly a suitable model when the data manifold appears due to simple transformations (e.g. rotations) of data. This is also exactly the type of data considered in the paper.
*) In Eq. 3, should it be "expm" instead of "exp" ?
*) The first two paragraphs of Sec. 2 are background material, whereas paragraph 3 and beyond describe material that is key to the paper. I would recommend introducing a \subsection (or something like it) to make this more clear.
*) The idea of working with transformations of data rather than the actual data is the corner-stone of Ulf Grenander's renowned "Pattern Theory". A citation to this seminal work would be appropriate.
*) In the first paragraph of the introduction links are drawn to the neuroscience literature; it would be appropriate to cite a suitable publication.

Pros(+) & Cons(-):
+ Well-written.
+ Good illustrative experiments.
- Real-life experiments are lacking.
- Limited methodology contribution.
- The assumed dynamics might be too simplistic (at least a discussing of this is missing).

For the AC:
The submitted paper acknowledges several grants (including grant numbers), which can directly be tied to the authors identity. This may be a violation of the double blind review policy. I did not use this information to determine the authors identity, though, so this review is still double blind.

Post-rebuttal comments:
The paper has improved with the incorporated revisions, but my main concerns remain. I find the Swiss Roll / rotated-USPS examples to be too contrived as the dynamics are exactly tailored to the linear ODE assumption. These are examples where the model assumptions are perfect. What is unclear is how the model behaves when the linear ODE assumption is not-quite-correct-but-also-not-totally-incorrect, i.e. how the model behaves in real life. I didn't get that from the newly added experiment. So, I'll keep my rating as is.

---

### Official Review · AnonReviewer2 · 2017-11-24
**Learning transport operators**

**Rating:** 5
**Confidence:** 4

**Review:**

This paper propose to learn manifold transport operators via a dictionary learning framework that alternatively optimize a dictionary of transformations and coefficients defining the transformation between random pairs of data points. Experiments on the swiss roll and synthetic rotated images on USPS digits show that the proposed method could learn useful transformations on the data manifold.

However, the experiments in the paper is weak. As the paper mentioned, manifold learning algorithms tend to be quite sensitive to the quality of data, usually requiring dense data at each local neighborhood to successfully learn the manifold well. However, this paper, claiming to be learn more rubust representations, lacks solid supporting experiments. The swiss roll is a very simple synthetic dataset. The USPS is also simple, and the manifold learning is performed on synthetic (rotated) USPS digits with only 1 manifold dimension. I would recommend testing the proposed algorithm on more complicated datasets (e.g. Imagenet or even CIFAR images) to see how well it performs in practice, in order to provide stronger empirical supports for the proposed method. At the current state, I don't think it is good for publishing at ICLR.

=========================
Post-rebuttal comments

Thanks for the updates of the paper and added experiments. I think the paper has improved over the previous version and I have updated my score.

---

### Official Review · AnonReviewer1 · 2017-11-27
**Interesting paper on learning transport operators; but somewhat confusingly written with unclear novelty.**

**Rating:** 4
**Confidence:** 4

**Review:**

Summary:

The paper considers the framework of manifold transport operator learning of Culpepper and Olshausen (2009), and interpret it as obtaining a MAP estimate under a probabilistic generative model. Motivated by this interpretation, the authors propose a new similarity metric between data points, which leads to a new manifold embedding method. This also leads the authors to propose a new transfer learning mechanism that can lead to improvements in classification accuracy.
Some representative simulation results are included to demonstrate the efficacy of the proposed methods.

Main comments:

This direction is interesting.  But unfortunately the paper is confusingly written and several points are never made clear. The conveyed impression is that the proposed methods are mainly incremental additions to the framework of Culpepper and Olshausen.

It would be far more helpful if the authors would have clearly described the following in more detail:
-  The new manifold embedding algorithm in Section 2 --  a proper explanation of the similarity measure, what the role of the MSE is in this algorithm, how to choose the parameters gamma and zeta etc.
- Why the authors claim that this method is more robust than other classical manifold learning methods. There certainly seems to be some robustness improvement over Isomap -- but this is a somewhat weak strawman since Isomap is notoriously prone to improper neighborhood selection.
- Why the transport operator viewpoint is an improvement over other out-of-sample approaches in manifold learning.
- Why the data augmentation using learned transport operators would be more beneficial than augmentation using other mechanisms (manual rotations, other generative models).

etc.


Other comments/questions:

- Bit confused about the experiment for Figure 1. Why set gamma = 0? Also, you seem to be fixing the number of dictionary elements to two (suggesting an ell-0 constraint), but also impose an ell-1 constraint. Why both?
- From what distribution are the random coefficients governing the transport operators drawn (uniform? gaussian?) how to choose the anchor points?
- The experiment in USPS digits is somewhat confusing. Rotations are easy to generate, so the "true rotation" curve is probably the easiest to implement and also the best performing -- so why go through the transport operator training process at all?  In any case, I would be careful to not draw too many conclusions from a single experiment on MNIST.

================

Post-rebuttal comments:

Thanks for the response. Still not convinced, unfortunately. I would go back to the classification example: it is unclear what the benefits of the transport operator viewpoint is over simply augmenting the dataset using rotations (or "true rotations" as you call them), or translations, or some other well-known parametric family. Even for the faces dataset, it seems that the transformations to model "happiness" or "sadness" are fairly simple to model and one does not need to solve a complicated sparse regression problem to guess the basis elements.  Consider fleshing this angle out a bit more in detail with some more compelling evidence (perhaps test on a bigger/more complex dataset?).

---

### Decision · Program_Chairs · 2018-01-29
**ICLR 2018 Conference Acceptance Decision**

**Decision:**

Reject

**Comment:**

Learning identity-preserving transformations from unlabeled data is definitely an important and useful direction. However the paper does not have convincing experiments to establish the effectiveness of the proposed method on real datasets which is a crucial limitation in my view, given that the paper is largely based on an earlier published work by Culpepper and Olshausen (2009).